# Tracking the Progression of the Simulated Bronze Disease—A Laboratory X-ray Microtomography Study

**DOI:** 10.3390/molecules28134933

**Published:** 2023-06-23

**Authors:** Zedong Wang, Xiaoqi Xi, Lei Li, Zhicun Zhang, Yu Han, Xinguang Wang, Zhaoying Sun, Hongfeng Zhao, Ning Yuan, Huimin Li, Bin Yan, Jiachang Chen

**Affiliations:** 1Henan Key Laboratory of Imaging and Intelligent Processing, PLA Strategic Support Force Information Engineering University, Zhengzhou 450001, China; miti_wang@163.com (Z.W.);; 2Henan Provincial Institute of Cultural Relics and Archaeology, Zhengzhou 450001, China

**Keywords:** bronze disease, in situ X-ray microtomography image, simulation experiment, metal corrosion

## Abstract

The internal three-dimensional characteristics of X-ray microtomography (micro-CT) has great application potential in the field of bronze corrosion. This work presents a method of simulating bronze disease based on an in situ micro-CT image to study the characteristics of the oxidative hydrolysis reactions of copper(I) chloride and copper(II) chloride dihydrate. A series of high-resolution reconstruction images were obtained by carrying out micro-CT at three key points throughout the experiment. We found that the reactions of copper(I) chloride and copper(II) chloride dihydrate showed different characteristics at different stages of the simulation in the micro-CT view. The method proposed in this work specifically simulated one single type of bronze corrosion and characterized the evolution characteristics of simulated bronze disease. It provides a new perspective to investigate bronze disease and can help improve the subsequent use of micro-CT to distinguish real bronze corrosions.

## 1. Introduction

Bronze disease, the most serious corrosion in various types of patinas, is caused by cyclic reactions of nantokite with the internal and external bronzes under the influence of oxygen and water [1]. The visual effect of bronze disease is the formation of pale green, powdery spots of copper trihydroxychlorides and is also accompanied by the stress cracking of the surface due to the morphology change and subsequent deeper penetration of nantokite into the bronzes [2]. Generally, the typical corrosion products on bronze disease in museum environments are shown in Table 1.

The corrosion of bronzes is a very complex process, which is influenced not only by environmental factors (burial, atmosphere, interior, sea) which cause different forms of bronze corrosion, but also by its own factors (different types of alloys) that cause different forms of corrosion reactions (chemical corrosion, electrochemical corrosion) and produce corrosion products with different characteristics: some bronze corrosion products have a protective effect on the object (cuprite, kenorite, malachite, langite, etc.), which should be retained as much as possible during the object’s cleaning process; some bronze corrosion products will further corrode the object’s body under certain conditions (nantokite, atacamite, clinoatacamite, etc.), which should be removed during the cleaning process according to the corrosion condition [1]. Therefore, the different corrosion products of bronze are treated differently in the cleaning process due to their different chemical properties, on the one hand, the harmful patina should be removed to avoid long-term damage, and on the other hand, the harmless patina should be retained to preserve the value of the bronzes. Nowadays, mechanical cleaning based on the type of bronze corrosion is becoming one of the main methods of cleaning bronze, and it is very important to have an accurate determination before cleaning the bronze.

Past studies on bronze corrosion have shown that corroded bronze surfaces have been monitored to investigate the burial environment, the composition of corrosive materials and the microstructure of corroded surfaces via various scientific analytical techniques, such as optical microscopy (OM), X-ray diffraction (XRD), scanning electron microscopy (SEM), dispersive energy spectroscopy (EDS) and Raman spectroscopy (RS) [4,5,6,7]. These scientific analytical techniques can effectively identify the corrosion products on bronze surfaces. However, only relying on the aforementioned analytical methods makes it difficult to obtain an accurate image of the potential corrosion state within the bronze without damage and carry out an accurate bronze corrosion cleaning program. Due to this problem, X-ray microtomography, which has excellent nondestructive internal detection capabilities, can aid in the study of the distribution and the three-dimensional damage of bronze disease within the bronze [8,9], and we believe that the use of micro-CT will facilitate the cleaning and protection of the bronze.

More recently, micro-CT has been progressively applied to the inspection of bronze [10,11,12,13,14,15,16,17], but most of these have always been limited to the simple scanning observations of objects with less exploration of the internal characteristics and corrosion of the objects. Due to the limitations of micro-CT imaging principles and the inadequate micro-CT study of a single bronze corrosion type, achieving the accurate differentiation of bronze corrosion types by micro-CT is difficult. However, using the in situ micro-CT experiment and the simulation experiment to distinguish the complex corrosion types on the bronze surface, we can observe the formation process of a single bronze corrosion type independently and obtain its unique structural characteristics via micro-CT.

The in situ micro-CT experiment provides a comprehensive view of the evolution of experimental specimens, which has a good application prospect for studying the evolution of various types of bronze corrosion. Although the in situ micro-CT experiment has not been carried out in the field of bronze examination, the use of micro-CT is well established. Studies in the field of metal corrosion have begun to use the in situ micro-CT experiment to investigate the internal damage and durability changes caused by material corrosion, and the in situ micro-CT experiment has significantly contributed to the development of the field [18,19]. In situ micro-CT experiment has the potential for a more in-depth study on bronze corrosion.

The simulation experimental method enables a more targeted study of a single type of bronze corrosion because it can design a reasonable experimental procedure according to the principle of bronze corrosion, select a reasonable corrosion starting point and eliminate the interference of other bronze corrosion [20,21,22,23,24]. The simulation experimental method proposed in this article is an efficient implementation of an exploratory micro-CT imaging process to study the characteristics of the formation of bronze disease corrosion products, which is the most damaging type of bronze corrosion. The key objectives of this paper are: (i) designing an experimental program that is consistent with the principles of bronze disease production without introducing other bronze corrosion reactions; (ii) designing the micro-CT experiment in accordance with the process of bronze corrosion; and (iii) demonstrating the correctness of the simulation experiment is essential.

This study emphasizes the advantages of using micro-CT for investigating the high-resolution three-dimensional nondestructive characterization of bronze disease and the in situ simulation method for studying evolutionary processes and characteristics’ changes due to bronze disease.

## 2. Experimental Methods

### 2.1. Corrosive Specimen and Environment

#### 2.1.1. Corrosion Specimen Preparation

Past study has shown that the nantokite layer (copper(I) chloride) is the starting point of bronze disease [25], so in order to produce an obvious experimental phenomenon, we regard the initial simulation experimental object as the stable bronze surface in the museum which was already corroded and had the nantokite layer. In conjunction with previous studies [1,4,26,27,28,29] on bronze disease and the multi-layered corrosion structure of bronzes, while in principle not introducing impurities that are not related to the formation of bronze diseases, we chose six artificial powders to prepare the experimental specimen, all of which were purchased from Tianjin Kemiou Chemical Reagent Co (China): copper (Cu, 99.5% purity), copper(I) chloride (CuCl, 97.0% purity), copper(I) oxide (Cu_2_O, 97.0% purity), calcium sulfate dihydrate (CaSO_4_·2H_2_O, 99.0% purity), copper(II) carbonate hydroxide (Cu_2_(OH)_2_CO_3_, 99.0% purity) and copper(II) chloride dihydrate (CuCl_2_·2H_2_O, 99.0% purity).

Among the six artificial powders, copper powder, copper(I) chloride, copper(I) oxide and copper(II) carbonate hydroxide form the basic structure of the experimental specimen which is consistent with Scott’s [1] summary of the multilayer corrosion structure. On this basis, the copper(II) chloride dihydrate layer was added over the cupric carbonate basic to study the effect of bronze disease on the stable corrosion product (malachite, Cu_2_(OH)_2_CO_3_). As chloride ions are the source of bronze disease [30], the use of copper(II) chloride instead of copper(I) chloride would not affect the production of bronze disease and would better simulate the appropriate oxygen environment for the secondary corrosion products on the bronze surface. In addition, calcium sulfate dihydrate was used to isolate the oxidative hydrolysis reaction of the two different reactants in the specimen. Figure 1 shows the schematic and physical drawings of the specimen with the corresponding geometrical dimensions.

In contrast to previous simulation experiments investigating bronze corrosion, this study used copper instead of bronze alloy, because the Pb, Sn and Zn elements in the bronze alloy will instigate many complex chemical reactions during the corrosion process and affect the imaging results of the micro-CT, SEM and other inspection techniques [31,32]. As the simulation experiment is aimed at studying bronze disease, we do not want the corrosion phenomena of other impurity metals to affect the micro-CT observations of bronze disease. In addition, we chose copper powder and other metal compound powders to carry out the simulation experiment to better simulate the fragile multilayer structure of the bronze surface [33] and the irregular copper matrix, and using copper powder will accelerate the simulation of bronze disease corrosion, provides a better view of the characterization changes caused by the bronze disease and will not have a significant impact on the authenticity of the simulation.

#### 2.1.2. Corrosive Environment Setup

The corrosive environment setup is shown in Figure 2. To achieve a more realistic simulation of a museum environment in which bronzes are stored, the corrosive environment is maintained at a constant temperature (20 °C) and humidity (50%) during the simulation experiment, which is following the standards of the bronze museum.

#### 2.1.3. Experimental Principle and Expected Phenomenon

In a marine or buried environment, chloride ions can react with copper to produce a layer of copper(I) chloride located at the interface between the external corrosion products and the surviving core metal matrix [34]. Once excavated, the copper(I) chloride layer is the starting point for bronze disease due to the changing environment, namely elevated oxygen levels [25]. Thus, in the experimental specimen, the first reaction to occur is the oxidative and hydrolysis of copper(I) chloride (Equation (1)) [1,27]. After that, hydrogen ions and chloride ions generated by Equation (1) further react with copper(I) oxide in the upper interfaces of the copper(I) chloride layer (Equation (2)). As the cupric ions in the environment increase, it will disturb the equilibrium between the metallic copper and the cuprous and cupric ions, promoting Equation (3) in the lower interfaces of the copper(I) chloride layer, and this will produce more copper(I) chloride and promote Equation (1) to proceed in the positive direction [35,36]. Figure 3a shows the evolution of the expected corrosion region in the upper and lower interfaces of the copper(I) chloride layer as the simulation experiment progresses. Independent corrosion layers are produced between the two different material layers in the specimen. Figure 3b summarizes the rationale for the above-expected phenomena based on Equations (1)–(3). The above reactions constitute the basic principle of cyclic copper corrosion of copper(I) chloride, which can basically explain the occurrence of bronze disease.
(1)$$4\mathrm{CuCl}+O_{2}\;+4H_{2}O\;\to 2Cu_{2}(OH{)}_{3}\mathrm{Cl}\;+\;2H^{+}\;+\;2{\mathrm{Cl}}^{-}$$
(2)$${\mathrm{Cu}}_{2}O+2H^{+}+2{\mathrm{Cl}}^{-}\to \;2\mathrm{CuCl}\;+\;2H_{2}O$$
(3)$$\mathrm{Cu}+{\mathrm{Cu}}^{2+}\to 2{\mathrm{Cu}}^{+}$$

Since chloride ions are the source of bronze disease [30], it is theoretically feasible to begin the simulation experiment using copper(II) chloride dihydrate as a substitute for copper(I) chloride [1]. However, the factors that control the conditions for the formation of atacamite or clinoatacamite are subtle. In contrast to the oxidation hydrolysis of copper(I) chloride to produce clinoatacamite, copper(II) chloride dihydrate will produce a mixture of atacamite and clinoatacamite, with more atacamite [37]. Therefore, due to the different percentages of the corrosion products, the characterization of the corrosion layer created by copper(II) chloride dihydrate will potentially be different. The specific characterization changes of copper(I) chloride and copper(II) chloride dihydrate are explored using micro-CT, and in the analysis of the micro-CT images, the characterization changes in the simulation experiment specimen produced by copper(I) chloride and copper(II) chloride dihydrate are discussed separately.

### 2.2. Experimental Process Detection and Product Analysis

#### 2.2.1. X-ray Microtomography Experiment

The micro-CT experiments were performed a total of five times (initial state, one week later, one month later, two months later and four months later) to ensure that the images of the specimen changes at each stage were captured. The micro-CT experiment was conducted with a ZEISS Xradia 510 Versa, which enables comparative observation of a wide range of materials and has good contrast to different materials [38]; the micro-CT system consisted of an X-ray source, a sample stage, a CCD camera and a detector (Figure 4). The X-ray energy and the exposure time affect the quality of the images, and the settings of the CT scan parameters are closely related to the nature of the material to be measured [39,40,41]; therefore, the micro-CT parameters were as follows: X-ray source, 140 kV/71 μA; pixel size, 32 μm; voxel size, 32 × 32 × 32μm; rotation angle, 180 rotation angle; and exposure time, 3 s. To ensure that the scanning results of different stages have a better lateral comparison, the same scanning parameters were maintained for each micro-CT experiment.

In order to better discuss the characteristic changes of the specimen at different stages in micro-CT images from a quantitative perspective, based on the traditional CT value, we defined an H_Cu_ value calculation method relative to the copper’s attenuation coefficient to characterize the density changes of each material layer in the specimen, as shown below:(4)$$H_{Cu}=1000\times \frac{{\mu -\mu }_{Cu}}{\mu_{Cu}}$$
where μ and μ_Cu_ are the attenuation coefficients for matrix material and the copper in the specimen. For noninvasive CT images (the contrast mechanism of CT is dominated by X-ray attenuation), combined with the Lambert−Beer law [42], the gray value intensities I reconstructed in the tomograms are approximately proportional to the attenuation coefficient μ according to [11], so the attenuation coefficient in Equation (4) can be approximately expressed by the gray value I in the micro-CT image:(5)$$H_{Cu}=1000\times \frac{I-I_{Cu}}{I_{Cu}}$$
where I and I_Cu_ are the gray value intensities for the matrix material and the copper in the specimen. From the definition of Equation (5), H_Cu_ for air and copper are −1000 and 0, and if the H_Cu_ value of the material is relatively large, the material usually has a high density.

#### 2.2.2. Detection of Corrosion Products by XRD Analysis and OM Observation

The XRD of the corrosion products generated later in the simulation experiment was carried out using a Bruker XRD (Bruker D8 advance diffractometer, Bruker Germany) to verify that the product of the simulation experiment was copper trihydroxychloride, the major component of bronze disease. The XRD parameters were as follows: tube current, 40 mA; tube voltage, 40 kV; scanning range, 5–70°; scanning step, 0.02°.

After the XRD analysis, a Zeiss Axio Imager.M2m microscope (Carl Zeiss, Oberkochen, Germany) with Zen core v3.1 software was used for optical visualization.

## 3. Results and Discussion

### 3.1. Corrosion of the Specimen by Using Micro-CT

Since two adjacent micro-CT datasets from the early simulation experiment varied only slightly, for simplicity, we selected three datasets (initial state, one month later, four months later) from a total of five for further analysis, of which three datasets are defined as the three different stages of the simulation experiment. Figure 5a shows the 3D reconstructions and the X-Y slices of the specimens in the three micro-CT datasets.

In the reconstructed micro-CT image, materials with higher X-ray absorption appear in the brighter regions and higher grayscale values, while materials with low X-ray absorption appear in the reconstructed micro-CT image as darker regions and have smaller grayscale values, we can observe that an evenly distributed single material layer shows a small variation in the greyscale value in the micro-CT image and that significant differences exist between different material layers in Figure 5a. Compared to the other compounds in the specimen, the copper powder has the highest attenuation, and as a result, the bottom copper powder appears to be a brighter area in the reconstructed micro-CT image. Comparing the X-Y slices of the specimen from the three different stages, grayscale value changes are observed at the interface between the copper(I) chloride layer and the copper(II) chloride dihydrate layer with other copper compounds. The volume of the specimen is significantly elevated by the formation of the corrosion layers.

Figure 5b shows the comparison of the greyscale curve along the three stages of the X-Y slices acquired in Figure 5a, and the local images (Figure 5(b1–b3)) corresponding to the areas of significant change in the greyscale value curve are listed for comparison. In Figure 5(b1), the slice images of the area near the interface between copper(II) chloride dihydrate and copper(II) carbonate hydroxide from the three experimental stages were compared; these results showed that the copper(II) carbonate hydroxide powders near the copper(II) chloride dihydrate layer turned darker in the reconstructed micro-CT image due to the corrosion reaction. Figure 5(b2,b3) show a low greyscale value and separate material layers proceeding at the upper and lower interfaces of the copper(I) chloride layer.

Table 2 shows the average H_Cu_ values of the original six materials in the specimen during the three stages of the simulation experiment. The average H_Cu_ values of the five compounds in the specimen are all less than zero, which indicated that the densities of the five compounds were all less than the density of copper. As the simulation experiment progresses, the average H_Cu_ values of the five compounds become larger, indicating that the densities of the five compounds become larger.

In addition, some of the copper(I) oxide and copper powders in the specimen are lumpy or granular, with a constant greyscale value during the whole corrosion experiment. Due to this phenomenon, we correlate the different stages of the micro-CT datasets to find the same location in the slice image. The subsequent simulation experimental analysis will focus on the reaction of copper(I) chloride, copper(I) oxide and copper, respectively, and the reaction of copper(II) chloride dihydrate and copper(II) carbonate hydroxide, depending on the phenomena shown in the X-Z slices of the three stages (Figure 6).

#### 3.1.1. Evolution of the Reaction among Copper(I) Chloride, Copper and Copper(I) Oxide

Figure 7a,b shows the upper and lower interfaces of the copper(I) chloride layer in the initial state of the specimen; a clear boundary between the two materials can be observed.

According to the X-Y section in Figure 8, there are two considerably thin corrosion layers on the upper and lower surfaces of the copper(I) chloride layer. Figure 8a shows the interface of the copper(I) chloride layer and the copper(I) oxide layer after one month. In the reconstructed micro-CT image, there is a thin dark layer of material produced between the two material layers inside the specimen, as highlighted by the white circle in the local image (the average H_Cu_ value is −512.4). In addition, Figure 8b shows the interface of the copper(I) chloride layer and the copper powder layer after one month. Compared to Figure 7b, a significant bulk of new dark production is observed at the boundary of the specimen (the average H_Cu_ value is −501.2). In combination with the optical images in Figure 8, the new corrosion products were verified to be produced on the upper and lower surfaces of the copper(I) chloride layer.

Four months after the simulation experiment, the independent corrosion layers formed inside the specimen, as shown in the X-Y slice of Figure 9. A comparison between Figure 9a,b shows that the new material layer formed by copper(I) chloride and copper(I) oxide contains a significant number of large pores and shows significant delamination (the average H_Cu_ value is −307.1), and the new material layer formed by copper(I) chloride and copper is relatively dense (the average H_Cu_ value is −296.7).

The in situ simulation experiment provides valuable insight into the corrosion mechanism and the corrosion characteristics of copper(I) chloride. Throughout the simulation experiment, the processes of corrosion reactions among copper(I) chloride, copper and copper(I) oxide are essentially the same as what was expected prior to the experiment: at the early stage of the simulation experiment, the corrosion reaction that occurred between copper(I) chloride and copper(I) oxide is relatively insignificant, but the micro-CT experiments successfully observed the occurrence of corrosion reactions; however, at the later stage of the corrosion experiment, the structural damage caused by copper(I) chloride and copper(I) oxide to the interior of the specimen is evident and severe, but the reaction product of copper(I) chloride and copper is not obvious in the specimen.

In addition, the corrosion layers on the upper and lower surfaces of the copper(I) chloride layer have almost similar average H_Cu_ values that the average H_Cu_ value of the corrosion layer between the copper(I) chloride layer and the copper powder layer is slightly larger, and the average H_Cu_ value, the corrosion layers obviously increases in the simulation experiment, which shows that the densities of the corrosion layers increase with the simulated corrosion reaction.

#### 3.1.2. Evolution of the Reaction between Copper(II) Chloride Dihydrate and Copper(II) Carbonate Hydroxide

In Figure 5a, the originally separated copper(II) carbonate hydroxide layer was gradually mixed with copper(II) chloride dihydrate. Furthermore, Figure 10 shows the detailed comparison of the interface between the copper(II) chloride dihydrate layer and the copper(II) carbonate hydroxide layer during the three stages: when the experiment was carried out for one month, each copper(II) chloride dihydrate reacted from its surface throughout the entire crystal and produced a thin dark layer (the H_Cu_ value is −580.8) on the surface in the reconstructed micro-CT image according to Figure 10b; when the experiment was carried out for four months, the copper(II) chloride dihydrate completely darkened in the reconstructed micro-CT image according to Figure 10c, and the copper(II) carbonate hydroxide layer surrounding copper(II) chloride dihydrate also showed a certain degree of corrosion, which caused a dark and loose corrosion layer in the reconstructed micro-CT image (the average H_Cu_ value is −445.8).

The changing average H_Cu_ value of the corrosion layer produced by copper(II) carbonate hydroxide shows that the density of the corrosion layer increases with the progress of the reaction, and when compared to the corrosion layer produced by copper(I) chloride, the average H_Cu_ value of the corrosion layer produced by copper(II) chloride dihydrate is less than that of copper(I) chloride, which means that the corrosion layer produced by copper(II) chloride dihydrate is less dense.

In summary, comparing the corrosion reaction above with the corrosion reaction in the copper(I) chloride layer, the corrosion reaction between copper(II) chloride dihydrate and copper(II) carbonate hydroxide not only caused the original dense material layer to become loose and low density but also resulted in a larger volume of corrosion. Our experiment simulated the corrosion reaction of copper(II) chloride dihydrate with malachite (Cu_2_(OH)_2_CO_3_), a common and harmless patina of bronze. This experiment suggests that stable copper(II) carbonate hydroxide patination in bronze is unsafe and should also be an important conservation target in the conservation proposal. Based on the results obtained, it may be stated that the examined specimen offered the opportunity to investigate the characteristics of bronze disease, which are worth studying in depth to distinguish bronze disease from complex bronze alloy corrosion.

### 3.2. Analysis of the Corrosion Layer Using XRD and OM

Four months after the simulation experiment, XRD analysis was performed on the three new corrosion layers produced in the specimen to determine the specific corrosion product type after the corrosion reaction in the specimen and to verify the correctness of the experimental simulation of bronze disease. Figure 11 shows the XRD detection positions of three new substances and their corresponding detection results.

The XRD patterns of the three substances are compared with standard PDF cards. The results show that the main body of the three substances is a mixture of different percentages of clinoatacamite and atacamite, which are the main component of bronze disease. It can be also found that there are few diffraction peaks of other impurities in the XRD pattern, indicating that no other types of bronze corrosion occurred in the simulation.

In addition, from the optical image in Figure 11, it can be observed that copper(II) chloride dihydrate has an obvious corrosive effect on the copper(II) carbonate hydroxide, and the original dense powdered copper(II) carbonate hydroxide turned into the loose large-volume corrosion product. On the upper and lower surfaces of the copper(I) chloride layer, the corrosion layer between the copper(I) chloride layer and the copper(I) oxide layer is thicker than that between the copper(I) chloride layer and the copper powder layer.

In order to further compare the characteristics of the simulation corrosion products, and verify that the simulation experiment can correctly reflect the real bronze corrosion, we removed all the compounds from the simulation samples at the end of the simulation experiment. Three simulation corrosion products (substances 1, 2, 3) and the altered copper(I) chloride were observed by the OM method (Figure 12).

According to Figure 12, the microscopic image is different in color from the optical image in Figure 11a: the simulation corrosion products (substances 1, 2, 3) are both pale green in the optical image but blue in the microscopic images (Figure 12a,c,d); altered cuprous(I) chloride is yellow-green in the optical image but pale green in the microscopic image (Figure 12b). Past studies using OM to detect real bronze corrosion were compared, such as He Ling’s OM examination of corroded bronze Ding from the Yin Ruins in China, in which copper trihydroxychlorides were detected on the corroded surface of bronze using OM, XRD and SEM [6]. The colors of the simulation corrosion products in the microscopic images also appear in the microscopic observation of real bronze corrosion.

The results of compositional XRD analysis and OM observation are essentially the same as what was expected prior to the experiment. The three simulation corrosion products consist of different proportions of atacamite and clinoatacamite based on XRD analysis, but comparing the characteristic differences of corrosion reactions between copper(I) chloride and copper(II) chloride dihydrate in Figure 8, Figure 9 and Figure 10, we observed a large difference in the formation of corrosion product, which was potentially caused by the different percentages of copper complexes in the environment. It is known from past studies that the formation of atacamite and clinoatacamite is controlled by a series of competing steps [1,43,44,45,46]: clinoatacamite is more likely to form when a high coordination numbers of copper complexes are available in the environment; atacamite is more likely to form when the percentage of CuCl+ is high [37,47]. According to the OM observations, the three simulated corrosion products are similar in color and all are in powder form, which is consistent with the characteristics of realistic bronze disease productfigure.

## 4. Conclusions

In this study, we carried out a high-resolution three-dimensional nondestructive characterization of simulated bronze disease in a simulated museum environment using the simulation experiment and in situ micro-CT. Based on the micro-CT results, the following conclusions can be drawn:

The corrosion characterization of the corrosion products of copper(I) chloride and copper(II) chloride dihydrate are significantly affected by the type of reactants. For the corrosion reaction of copper(I) chloride: in the initial stage of the simulated bronze corrosion reaction, the characteristics of the corrosion reaction are two considerably thin layers; in the later stage of the reaction, the reaction between copper(I) chloride and copper produced a thick and dense corrosion layer, while the reaction of copper(I) chloride with copper(I) oxide produced a thick corrosion layer with a large number of pores. For the corrosion reaction of copper(II) chloride dihydrate, not only does it cause the original dense material layer to become loose and low density, but it also results in a larger volume of corrosion compared to the corrosion reaction of cuprous chloride in the simulation experiment. Although the corrosion reaction products are the same according to the XRD analysis, the reactions of copper(I) chloride and copper(II) chloride dihydrate give rise to corrosion products with different processes, internal structures and thicknesses.

This work provides a basis for micro-CT studies on the characterization of bronze disease and can aid in the identification of bronze disease corrosion in a wide range of bronze corrosion types. By combining simulation experiments with in situ micro-CT experiments, we succeeded in observing two different states of simulated bronze disease corrosion. This study proves that the micro-CT has a good effect in detecting the internal characteristics of the simulation bronze disease corrosion. Additionally, as this study is a simple simulation of bronze disease based on the basic chemical principle, even though the XRD and OM experiments show that the simulations have some similar characteristics to real bronze disease corrosion, the simulation is still not fully comparable to real corrosion, so it cannot give a comprehensive view of the characteristics of bronze disease in real bronze. In future work, this study can be further extended to include the more realistic phenomena of bronze disease, such as studying real bronzes for in-site corrosion observation or introducing a more comprehensive multi-layer bronze corrosion structure in a simulation experiment.

## Figures and Tables

**Figure 1 molecules-28-04933-f001:**
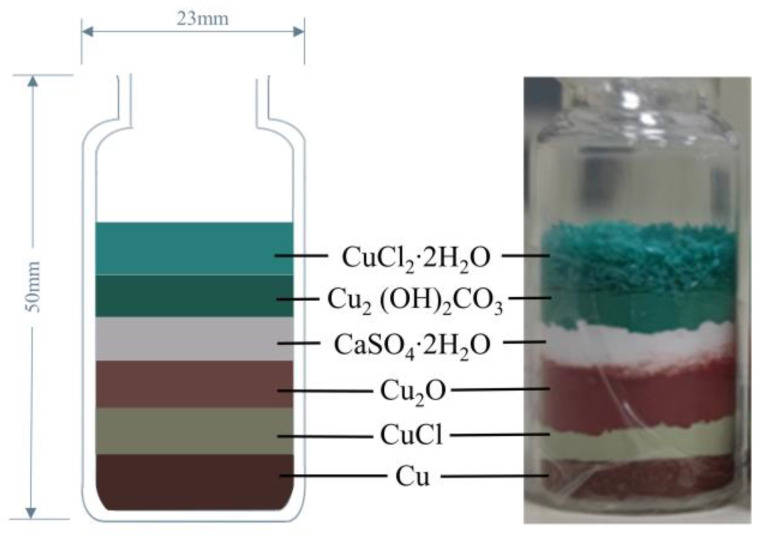
Schematic presentation of the specimen (each material layer filled approximately 5 mm in height).

**Figure 2 molecules-28-04933-f002:**
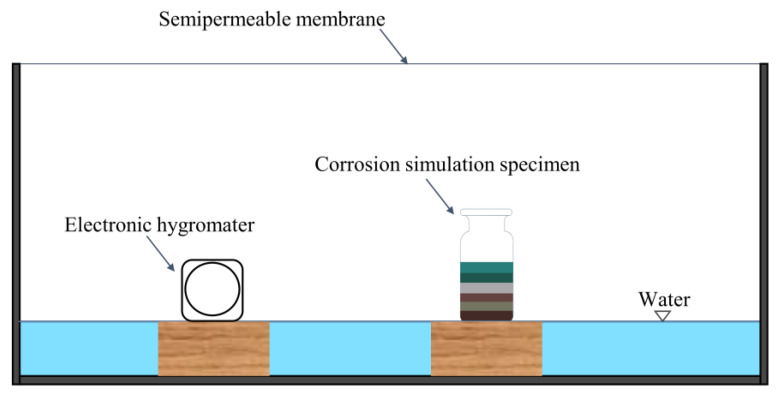
Corrosive environment setup for the simulation experiment. The function of semipermeable membranes is to ensure a high level of air humidity. During the experiment, we keep the humidity constant by replenishing the water in the environment and controlling the opening and closing of the semipermeable membrane.

**Figure 3 molecules-28-04933-f003:**
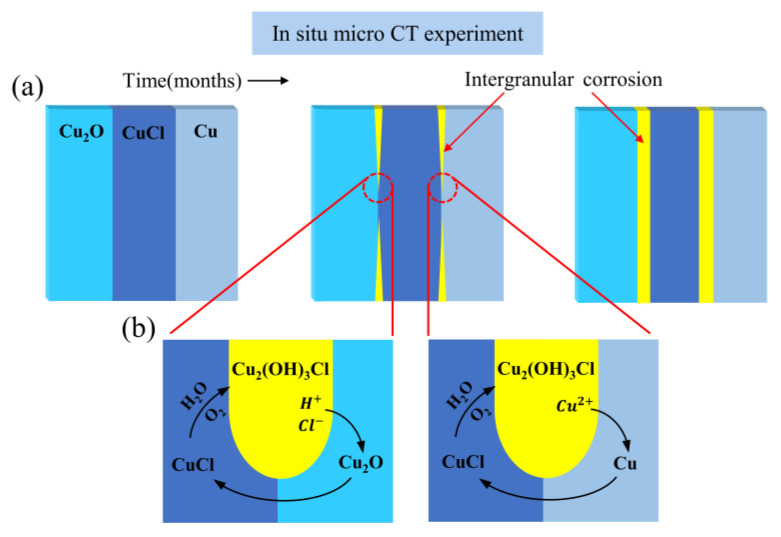
(**a**) Schematic representation of the expected corrosion evolution between the three material layers. Over the course of the simulation experiment, independent corrosion layers will be produced between the two different material layers in the specimen, and the main corrosion form of the corrosion layer is expected to be intergranular corrosion. (**b**) Schematic representation of the corrosion reaction principle among copper(I) chloride, copper and copper(I) oxide. The yellow area is the copper trihydroxychlorides produced by the corrosion reaction of copper(I) chloride.

**Figure 4 molecules-28-04933-f004:**
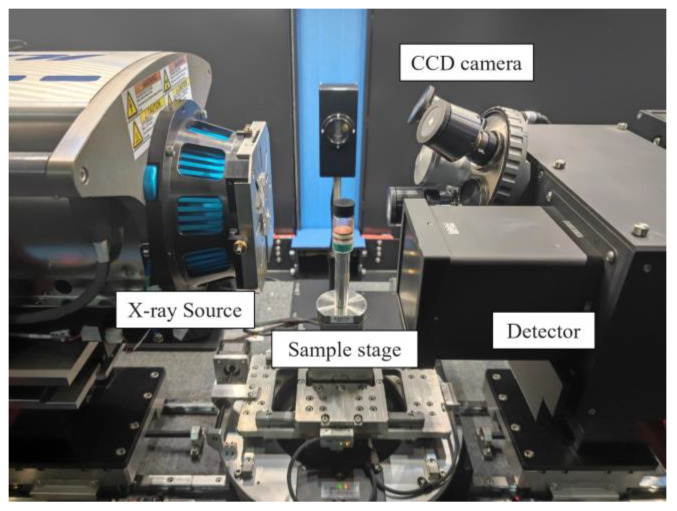
The micro-CT experiment for the specimen and the internal features of the ZEISS Xradia 510 Versa.

**Figure 5 molecules-28-04933-f005:**
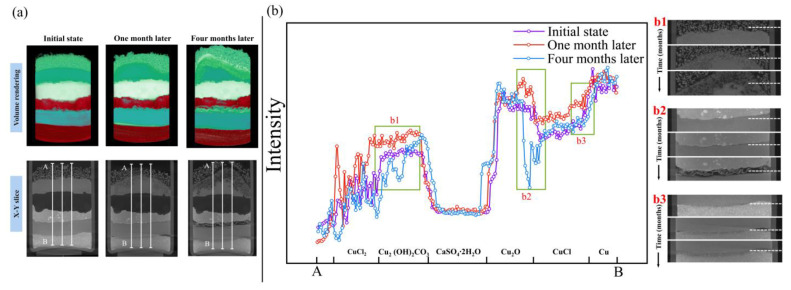
Data on the corrosion process. (**a**) The 3D visualization and reconstructed X-Y slices of the specimen for the three stages, and the scale of the images in the three stages are the same. The voxel greyscale values of the images along the white lines in the X-Y slices of the three experimental stages were compared and analyzed to determine the difference in greyscale values of the different material layers and the presence of the corroded region. (**b**) The results of the voxel greyscale value analysis show significant differences between the different material layers, and the three areas of the curve that show a large difference are marked in green boxes with comparisons of the local micro-CT slice images of the corresponding areas; these areas are the corroded regions (**b1**–**b3**). X-Y slice images visually compare the change in corrosion layer thickness at three stages.

**Figure 6 molecules-28-04933-f006:**
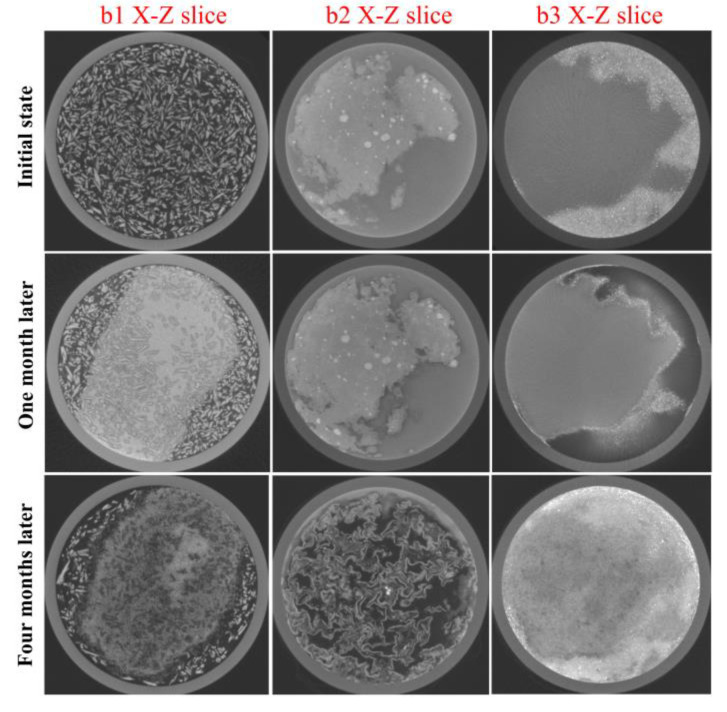
The complete internal images of corrosion layers, X-Z slices corresponding to the interfaces of b1, b2 and b3 (Figure 5b) for the three stages through the white dotted lines.

**Figure 7 molecules-28-04933-f007:**
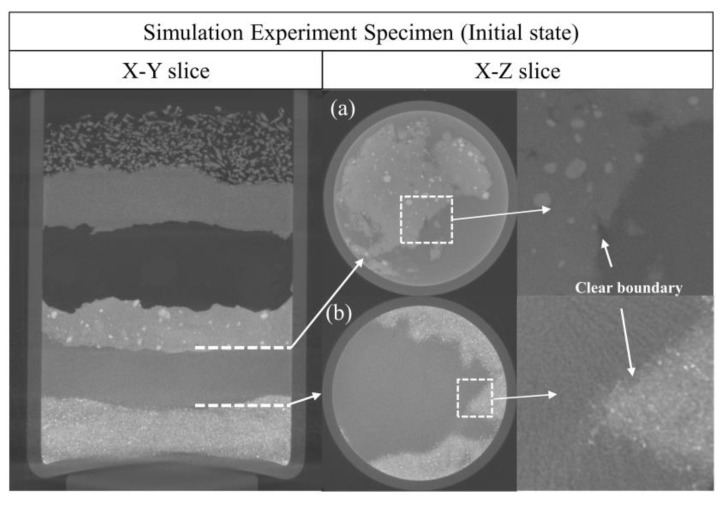
The upper and lower interfaces of the copper(I) chloride layer of the specimen are in the initial state. (**a**) The interface of the copper(I) chloride layer and the copper(I) oxide layer. (**b**) The interface of the copper(I) chloride layer and the copper powder layer. The Y coordinates of (**a**) and (**b**) are shown as the white dashed line in the X-Y slice image.

**Figure 8 molecules-28-04933-f008:**
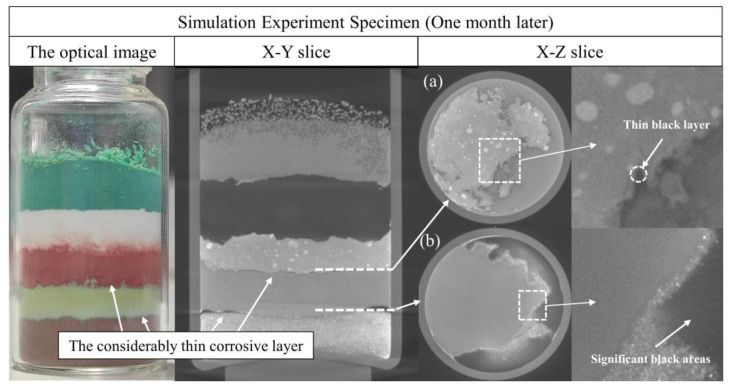
The upper and lower interfaces of the copper(I) chloride layer in the specimen after one month. (**a**) The interface of the copper(I) chloride layer and the copper(I) oxide layer. (**b**) The interface of the copper(I) chloride layer and the copper powder layer. The square regions in (**a**,**b**) show the black layer of material produced between the two material layers. The left image is the optical image of the specimen in the same stage, and two considerably thin blue-green corrosion layers appear on the upper and lower surfaces of the copper(I) chloride layer.

**Figure 9 molecules-28-04933-f009:**
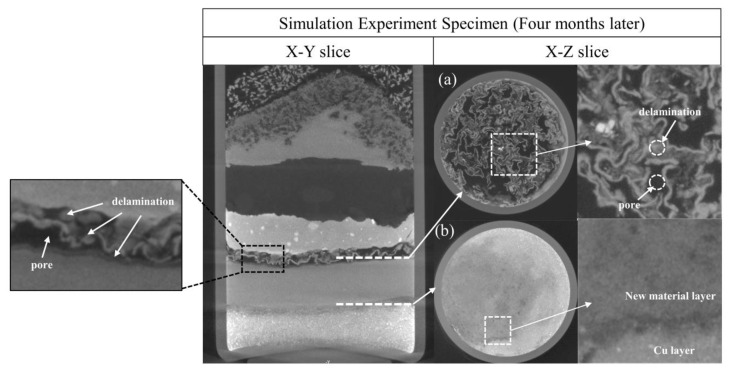
The upper and lower interfaces of the copper(I) chloride layer in the specimen after four months. (**a**) The interface of the copper(I) chloride layer and the copper(I) oxide layer. The square region in (**a**) shows the mixed material structure and extensive pores produced by the corrosion reaction, while the square region in the X-Y slice shows significant delamination. (**b**) The interface of the copper(I) chloride layer and the copper powder layer. The square region in (**b**) shows the significant greyscale difference between the new material layer and the copper layer.

**Figure 10 molecules-28-04933-f010:**
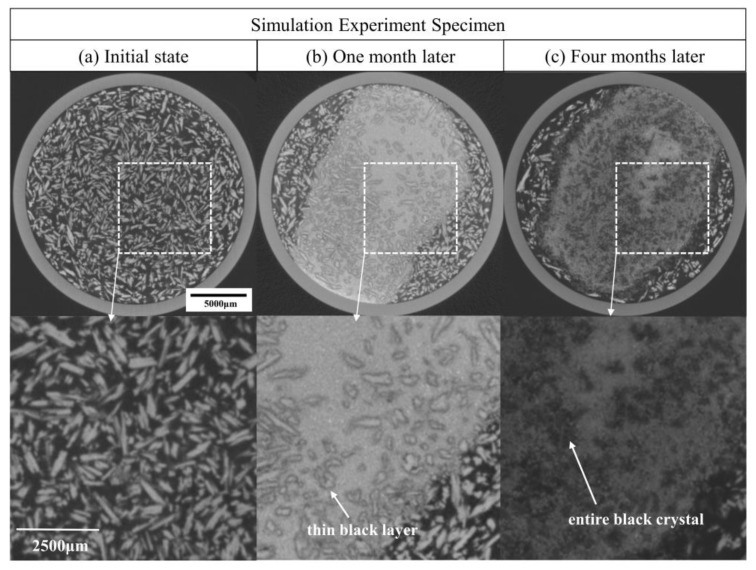
X-Z slice images of the specimen in the same position of the interface between the copper(II) chloride dihydrate layer and the copper(II) carbonate hydroxide layer for the three stages of the simulation.

**Figure 11 molecules-28-04933-f011:**
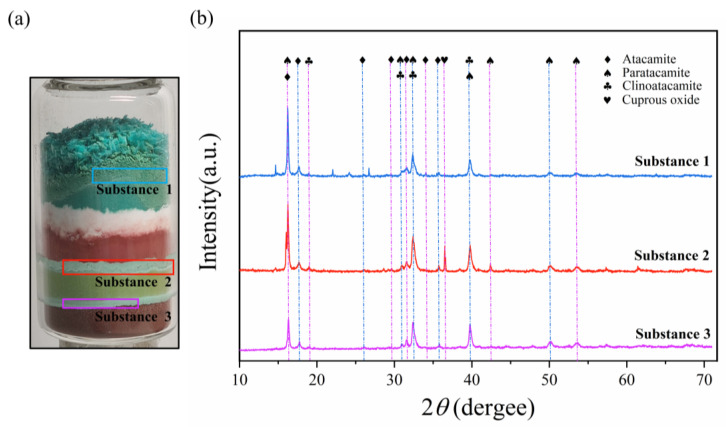
The XRD detection results of three new substances. (**a**) The optical image of the specimen after four months of the simulation experiment. (**b**) The XRD evaluation using diffractograms simulated from the PDF data. The different color curves correspond to the detection positions in the optical image through the corresponding color boxes. Since the term “clinoatacamite” has been widely accepted and can replace the term “paratacamite” [1], clinoatacamite and paratacamite can be regarded as the same substance.

**Figure 12 molecules-28-04933-f012:**
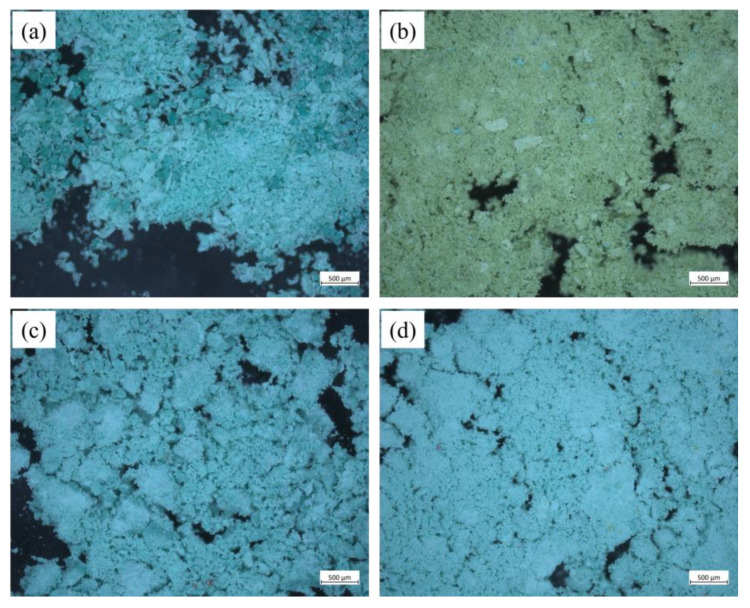
The optical microscopy detection results. (**a**) Substance 1 is produced by the copper(II) chloride dihydrate layer and the copper(II) carbonate hydroxide layer. (**b**) Altered copper(I) chloride at the end of the simulation. (**c**) Substance 2 is produced by the copper(I) oxide layer and the copper(I) chloride layer. (**d**) Substance 3 is produced by the copper(I) chloride layer and the copper powder layer.

**Table 1 molecules-28-04933-t001:** Characteristics of bronze disease-induced corrosion products [1,3].

Corrosion Product	Formula	Crystal System	Color
Nantokite	CuCl	Cubic	Pale green
Atacamite ^a^	Cu_2_(OH)_3_Cl	Orthorhombic	Pale green
Clinoatacamite ^a^	Cu_2_(OH)_3_Cl	Rhombohedral	Pale green

^a^ Atacamite and clinoatacamite are isomers of copper trihydroxychlorides.

**Table 2 molecules-28-04933-t002:** The average H_Cu_ values of the original six materials.

Experiment Stages	CuCl_2_·2H_2_O	Cu_2_(OH)_2_CO_3_	CaSO_4_·2H_2_O	Cu_2_O	CuCl	Cu
Initial state	−369.4	−416.5	−712.1	−165.8	−274.8	0
One month later	−302.7	−327.3	−690.9	−152.1	−231.1	0
Four months later	−212.1	−258.1	−644.8	−101.3	−211.9	0

## Data Availability

The data used for the manuscript are available for researchers upon request from the corresponding author.

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
