# Peer review of "Tracking the Progression of the Simulated Bronze Disease—A Laboratory X-ray Microtomography Study"

_molecules, 2023, doi:10.3390/molecules28134933_

Round 1
Reviewer 1 Report
This is an interesting paper on study of bronze corrosion. However, presents many weak points and, in my opinion, it should be better explained and reformulated.
The text is not always fluent and clear to read and some topics, in my opinion, need more investigation:
- Authors frequently refer to "corrosion morphology," which cannot be comparable to true corrosion morphology.
- The situation being studied is substantially different from the real one, both because the artefact's microstructure, which affects corrosion, is ignored and because the stratification, which simulates the patina of corrosion, is oversimplified.
- In a lab sample, the corrosion product layers have a thick layer, but in a real artefact, the layer thickness is frequently considerably thinner and there are no definite limits; might the patina imaging be affected?
- References should be checked.
The paper contains good pieces of work but can be published only after the suggested revisions.
Reviewer 2 Report
The present work presents an original idea about bronze disease, but its coherence, development and conclusions are very limited, mainly with regard to its applicability to real cases.
The title is quite long and confusing and does not clearly express the development of the work, and may even mislead the reader. The authors only use a simplified bronze analogue, oversimplifying the composition of the bronze objects they intend to characterize.
The abstract is very confusing and does not clearly show the results obtained, so it is not stimulating.
In the experimental part, there should be more consistency in the characterization of the compounds used (names, crystallinity, respective polymorphs, textural aspects). A typical case is the citation of copper trihydroxychlorides. I think that in general the nomenclature of the compound should be used and not the chemical composition, except when amorphous phases are formed.
Os parâmetros experimentais da XRD e Micro-CT são por vezes indicados incorretamente. Example XRD scanning step, rotation and spatial resolution.
The good visual data obtained by micro-CT show the need for further discussion of the results around the relative or absolute attenuation of X-rays (e.g. Hounsfield units (HU) are a dimensionless unit universally used in computed tomography (CT) scanning to express CT numbers in a standardized and convenient form).
The results obtained by micro-CT need better validation by optical observation or better characterization by XRD, in order to scale the conclusions to real situations.
Authors often refer to morphological variation as the object and contribution of this study, but in reality the morphology of reaction products is not fully appreciated. Eventually, optical visualization or SEM could help to appreciate the similarity with real cases or information already published.
Some more critical parts are indicated in yellow in the attached pdf, by way of example.

Need improvment in the nomenclature!
